# Antiproliferation Activity and Mechanism of c9, t11, c15-CLNA and t9, t11, c15-CLNA from *Lactobacillus plantarum* ZS2058 on Colon Cancer Cells

**DOI:** 10.3390/molecules25051225

**Published:** 2020-03-09

**Authors:** Qing Ren, Bo Yang, Guangzhen Zhu, Shunyu Wang, Chengli Fu, Hao Zhang, R. Paul Ross, Catherine Stanton, Haiqin Chen, Wei Chen

**Affiliations:** 1State Key Laboratory of Food Science and Technology, Jiangnan University, Wuxi 214122, China; renqing666@126.com (Q.R.); bo.yang@jiangnan.edu.cn (B.Y.); zhanghao@jiangnan.edu.cn (H.Z.); weichen@jiangnan.edu.cn (W.C.); 2School of Food Science and Technology, Jiangnan University, Wuxi 214122, China; zhugzname@163.com; 3Zhejiang Liziyuan Food Co., Ltd., Jinhua 321015, China; winnerwon@163.com (S.W.); fucl@liziyuan.com (C.F.); 4National Engineering Research Center for Functional Food, Jiangnan University, Wuxi 214122, China; 5Wuxi Translational Medicine Research Center and Jiangsu Translational Medicine Research Institute Wuxi Branch, Wuxi 214122, China; 6International Joint Research Center for Probiotics & Gut Health, Jiangnan University, Wuxi 214122, China; p.ross@ucc.ie (R.P.R.); catherine.stanton@teagasc.ie (C.S.); 7APC Microbiome Ireland, University College Cork, T12 K8AF Cork, Ireland; 8Teagasc Food Research Centre, Moorepark, Fermoy, P61 C996 Co. Cork, Ireland; 9Beijing Innovation Centre of Food Nutrition and Human Health, Beijing Technology and Business University (BTBU), Beijing 100048, China

**Keywords:** conjugated linolenic acid, caco-2 cell, lipid peroxidation, apoptosis, pyroptosis

## Abstract

Conjugated linolenic acid (CLNA) is a type of ω-3 fatty acid which has been proven to have a series of benefits. However, there is no study about the function of *Lactobacillus*-derived CLNA isomer. *Lactobacillus plantarum* ZS2058 has been proven to manifest comprehensive functions and can produce CLNA. To investigate the specific functions of CLNA produced by this probiotic bacterium, two different conjugated α-linolenic acid (CLNA) isomers were successfully isolated. These isoforms, CLNA1 (c9, t11, c15-CLNA, purity 97.48%) and CLNA2 (c9, t11, t15-CLNA, purity 99.00%), both showed the ability to inhibit the growth of three types of colon cancer cells in a time- and concentration-dependent manner. In addition, the expression of MDA in Caco-2 cells was increased by CLNA1 or CLNA2, which indicated that lipid peroxidation was related to the antiproliferation activity of CLNAs. An examination of the key protein of pyroptosis showed that CLNA1 induced the cleavage of caspase-1 and gasdermin-D, while CLNA2 induced the cleavage of caspase-4, 5 and gasdermin-D. The addition of relative inhibitors could alleviate the pyroptosis by CLNAs. CLNA1 and CLNA2 showed no effect on caspase-3, 7, 9 and PARP-1, which were key proteins associated with apoptosis. No sub-diploid apoptotic peak appeared in the result of PI single staining test. In conclusion, CLNA1 activated caspase-1 and induced Caco-2 cell pyroptosis, whereas CLNA2 induced pyroptosis through the caspase-4/5-mediated pathway. The inhibition of Caco-2 cells by the two isomers was not related to apoptosis. This is the first study on the function of *Lactobacillus*-derived CLNA isomer. The inhibition pathway of *Lactobacillus*-derived CLNA isomer on colon cancer cells were proved.

## 1. Introduction

Conjugated fatty acids (CFAs) are positional and geometric isomers of unsaturated fatty acids that contain one or more non-methylene interrupted double bonds in either the *cis* or *trans* forms [1,2,3]. One CFA that has been intensively investigated is conjugated linoleic acid (CLA). Numerous studies have reported that CLA exhibits a variety of health benefits, including anti-cancer [4,5], anti-oxidation [6], anti-atherosclerotic [7], anti-diabetic [8,9], and anti-obesity effects [10,11].

In addition to CLA, another group of conjugated fatty acids, the conjugated α-linolenic acid (CLNA) isomers, have recently received increased attention for having biological properties similar to those of CLA. Several plant-derived CLNAs have been studied, including punicic acid (PUA) from pomegranate and trichosanthes; α-eleostearic acid (α-ESA) and β-eleostearic acid (β-ESA) from tung, bitter ground and snake gourd seed; catalpic acid (CA) from catalpa, jacaric acid (JA) from jacaranda; α-calendic acid (α-CDA) and β-calendic acid (β-CDA) from pot marigold [12]. The chemical structure of plant-derived CLNA was shown in Figure 1.

The anti-cancer activity on cells of plant-derived CLNA isomers depends on the position and cis-trans configuration of the conjugated double bonds [4]. The descending order of CLNA isomer cytotoxicity on the colon cancer cell line HT-29 is trans9, trans11, trans13-CLNA followed by cis9, trans11, trans13-CLNA, then trans11, trans13-18:2, trans9, trans11-18:2, cis9, trans11-18:2 [13]. Further, β-eleostearic acid (t9, t11, t13-CLNA) and β-calendic acid (t8, t10, t12-CLNA), which have all-trans-conjugated double bonds, exerted stronger growth inhibition than α-eleostearic acid and α-calendic acid with the cis configuration in the human colon cancer cell line Caco-2 [14]. Based on the above reports, all-trans conjugated fatty acids may have more effective tumor suppressing activity. The lethal mechanism of plant-derived CLNAs has been extensively studied, mainly focusing on lipid peroxidation and induction of apoptosis [5,15]. Michael et al. found that cis9, trans11, cis13-CLNA can inhibit the growth of breast cancer cells through lipid peroxidation and the PKC pathway [16]. This cis9, trans11, trans13-CLNA can induce apoptosis in breast cancer cells by blocking the G2-M phase of mitosis, or by oxidative reactions [17]. The omega-3 fatty acid DHA has been reported to induce cancer cell pyroptosis; however, the interaction between pyroptosis and CLNAs—which are also a type of omega-3 fatty acid—is unknown.

CLNAs can also be extracted from microbial sources, and it could be that *Lactobacillus*-derived CLNAs and plant-derived CLNAs have different anticancer activities due to structural differences. With *Lactobacillus*-derived CLNAs, Coakley et al. and Hennessy et al. obtained a CLNA mixture (55.6% c9, t11, c15-CLNA) from bifidobacterial fermentation; this mixture inhibited SW480 cells with a lethality of 85% at a concentration of 180 μM, which is more toxic than both α-linolenic acid (ALA) and CLA [18]. Studies on the mechanism of action of this inhibitory activity indicate potential roles for increased cellular lipid peroxidation, altered cellular phospholipid composition, and reductions in the cellular content of Bcl-2 [19]. In our previous work, *Lactobacillus plantarum* ZS2058 was capable of isomerising α-linolenic acid into CLNA [20]. The structures of *Lactobacillus*-derived CLNA are c9, t11, c15-CLNA (CLNA1) and t9, t11, c15-CLNA (CLNA2), respectively, where the second or third double bond is in a different position compared with plant-derived CLNA. No studies have reported the efficacy of *Lactobacillus*-derived CLNA isomers, perhaps due to the difficulty of screening high-yield CLNA strains and the complexity of extracting and separating CLNA isomers. This study explores the cell-inhibitory effects of a bacterial-derived CLNA single isomer on colon cancer cells, evaluates the anti-cancer activities of conjugated linolenic acid, and provides a medical basis for the use of conjugated fatty acids as adjuvants in the treatment of cancer.

## 2. Results

### 2.1. Preparation and Purity Detection of CLNA1 and CLNA2

The α-linolenic acid was isomerized by *L. plantarum* ZS2058, and the CLNA mixtures were collected from the fermentation. The CLNA1 and CLNA2 were isolated by preparative liquid chromatography (Figure 2a) and their purity were detected by GC-MS. The purity of CLNA1 and CLNA2 reached 97.48% and 99.99% respectively. The mass spectrum and chemical structures of CLNA1 (c9, t11, c15-CLNA) and CLNA2 (c9, t11, t15-CLNA) are shown in Figure 2b, and the position of the double bond differed from that of plant-derived CLNAs (Figure 1).

### 2.2. Anti-proliferative Activity of CLNA1, CLNA2 Against Colon Cancer Cells

The anti-proliferative effects of CLNA isomers on three types of colon cancer cells were measured after incubation with the CLNA isomers for 24 h, 48 h, and 72 h, respectively. Both of the CLNA isomers exhibited antiproliferative activity on the three types of colon cancer cells in a dose-dependent manner, with a half maximal inhibitory concentration (IC_50_) ranging from 18.26 μM (for CLNA1 on Caco-2) to 67.52 μM (for CLNA2 on HT-29) after 48 h treatment (Figure 3a). Among the tested cell lines, Caco-2 cells were most inhibited, corresponding to an IC_50_ ranging from 18.26 to 19.75 μM, therefore, it was chosen for further investigation. CLNA1 and CLNA2 inhibited the proliferation of Caco-2 cells at concentrations of approximately 20 μM, whereas ALA, with an IC_50_ of 38.93 μM, had weaker antiproliferative activity (Table 1). Taking into consideration that cell death is characterized by typical morphological features, we analyzed the morphology of Caco-2 cells either untreated or treated with CLNAs at the IC_50_ for 24–48 h. The CLNAs treatment reduced the cell density and induced remarkable morphological changes. The majority of cells displayed an increase in size and evidence of cytoplasmic swelling, which are characteristic features of pyroptosis (Figure 3b).

To determine the possible mechanisms of the anti-proliferative effects of *Lactobacillus*-derived CLNAs, Annexin-V/PI staining was used to examine apoptosis in Caco-2 cells treated by CLNA1 or CLNA2. The proportion of cells in late apoptosis or necrosis increased with the concentration of CLNA1 added. At the IC_50_ of CLNA1, apoptosis had progressed to the middle and late stages, the early apoptosis rate of cells was 2.47%, and the rate of late apoptosis or necrosis was 23.5%. Treatment with CLNA2 (1/2 IC_50_-2 IC_50_) for 48 h led to a dose-dependent increase in the percentage of cells in late apoptosis, as demonstrated by Annexin V and PI double-positive cells. When the concentration of CLNA2 reached twice the IC_50_, the occupancy rate of living cells and early apoptosis was less than 10%, and most cells were in the late stages of apoptosis or necrosis (Figure 3c).

### 2.3. Induction of Lipid Peroxidation by CLNA1 and CLNA2 in Caco-2 Cells

To investigate the mechanism underlying the cytotoxic effects by *Lactobacillus*-derived CLNAs, Caco-2 cells were incubated with CLNAs and the inhibitor Vitamin E, which can block oxidative reactions. When the cells were treated with CLNAs in the presence of Vitamin E, the growth inhibition by CLNA1 and CLNA2 was attenuated, with the cells retaining 76% and 88% of their proliferation potential (Figure 4a). Therefore, the mechanism of the cytotoxicity of CLNA is likely to involve lipid peroxidation. Then the protein level of typical lipid peroxidation index MDA was determined. Increased expression levels of MDA were observed in CLNA-treated Caco-2 cells (Figure 4b). The expression level of MDA in the CLNA1 group reached its highest level at 24 h, while the MDA expression level in the CLNA2 group reached its highest level at 36 h, suggesting that Caco-2 cells have different reactions to the two different fatty acids. In summary, CLNA1 and CLNA2 stimulated Caco-2 cells to produce excessive MDA, which resulting in cell death. To investigate the mode of death in Caco-2 cells treated by *Lactobacillus*-derived CLNA, apoptosis, autophagy, necroptosis, and pyroptosis were investigated in the Caco-2 cell line.

### 2.4. CLNA1 and CLNA2 Induce Caco-2 Cell Death Independent of Apoptosis and Autophagy

To determine whether the mechanism of cell death from bacterial-derived CLNAs involves apoptosis, signaling genes and proteins known to be involved in apoptosis were analyzed by RT-qPCR (Figure 5a). PI single staining and Western blotting were also performed. There was no change in the mRNA expression of transcription factors PPARγ, TNF-α and TGF-β in Caco-2 cells (Figure 5b), which indicated that CLNA1 and CLNA2 did not play toxic role through the TGF-β/Smad3/IRS -1 pathway [21]. The plant-derived cis9, trans11, trans13-CLNA inhibits Caco-2 cells by up-regulating the transcriptional level of PPARγ [22]. Thus, the mechanisms of cell death between plant-derived and *Lactobacillus*-derived CLNA are distinct. In comparison, CLNA1 and CLNA2 cannot induce Caco-2 cell death by affecting the transcriptional levels of PPARγ, TNF-α, and TGF-β genes. In addition, similar tendencies were found in the expression levels of poly (ADP-ribose) polymerase 1 (PARP-1), caspase-3, caspase-7 and caspase-9 after Caco-2 cells were treated with CLNA1 and CLNA2 at different times and concentrations. The cleavage of PARP-1, caspase 3, caspase 7 and caspase 9 was not observed at any time point or concentration, which corresponds to the effect of the inhibitor Z-VAD (Figure 5d). These results amply demonstrated that CLNA1 and CLNA2 do not induce Caco-2 cell death through apoptosis.

A PI single staining test was used to determine whether *Lactobacillus*-derived CLNA affect the cell cycle in Caco-2 cells (Figure 5c). Because CLNA1 and CLNA2 at concentrations of twice the IC_50_ were highly toxic to Caco-2 cells after 48 h, we used the concentrations of half the IC_50_ and the IC_50_ to detect the cell cycle. We found that no sub-diploid apoptotic peak appeared and that no significant distinction in the proportion of Caco-2 cells in any cell cycle stage was observed with increase in CLNA1 and CLNA2 concentrations (Figure 5c). Thus, we conclude that CLNA1 and CLNA2 do not affect the cell cycle and induce Caco-2 cell apoptosis. In addition, other death mode inhibitors 3-MA, Nec-1s, and GSK-872 were not effective in relieving Caco-2 cell death from CLNA1. With CLNA2 treatment, we observed that cell death was enhanced when 3-MA was added. Similarly, the Caco-2 cell death induced by CLNA2 was slightly strengthened when RIPK3 was inhibited by the necroptosis inhibitor GSK-872. Nec-1s also had a significant effect on Caco-2 cells treated by CLNA2, unlike cells treated with CLNA1 (Figure 5d). Interestingly, the MLKL inhibitor NSA significantly inhibited the death of Caco-2 cells induced by CLNA1 or CLNA2, which indicates that the mode of CLNA action on Caco-2 cells may be related to the necroptosis pathway.

Taken together, the two CLNA isomers induce Caco-2 cell death by non-apoptotic or non-autophagic pathways, and the mechanism of death is likely to be related to the inhibitor NSA.

### 2.5. CLNA1 and CLNA2 Induce Caco-2 Pyroptosis by Different Pathways

To investigate whether the inhibitory mechanism of CLNA on Caco-2 cells is connected with pyroptosis, the key proteins known to be involved such as caspase-1, caspase-4, caspase-5, and GSDMD were assayed by Western Blotting. CLNA1 significantly increased the expression levels of cleaved caspase-1 and GSDMD compared with control (Figure 6a). To confirm that CLNA1-induced cell death was characterized by classical pyroptosis, the inhibitor VX-765 was applied, targeting caspase-1. VX-765 can inhibit the death of Caco-2 cells induced by CLNA1 (Figure 6b). Additionally, a Western blotting analysis of caspase-4 and caspase-5 confirmed that CLNA1 caused Caco-2 cell death not through the caspase 4/5 nonclassical pathway (Figure 6c).

The corresponding protein expression levels in Caco-2 cells treated with CLNA2 were different. Exposure to CLNA2 (IC_50_) significantly increased the expression of cleaved caspase-4, caspase-5, and GSDMD compared with controls (Figure 6c), but cleaved caspase-1 showed no difference compared with controls (Figure 6a), which is different from the results of CLNA1. To confirm that CLNA2-induced cell death was characterized by nonclassical pyroptosis, the caspase-4/5 inhibitor AC was applied. The AC inhibitor significantly inhibited the death of Caco-2 cells caused by CLNA treatment (Figure 6d).

## 3. Discussion

Probiotics have been claimed to possess functions such as suppressing inflammation, protecting the intestinal barrier, and increasing the body’s antioxidant capacity [23,24]. In some cases, the benefits of probiotics have been attributed to certain beneficial metabolites produced by them. In this study, we used the *Lactobacillus plantarum* ZS2058 to isomerise α-linolenic acid and assess the function of the CLNA produced. The structure of *Lactobacillus*-derived CLNA is different from that of plant-derived CLNA, and they can be defined as a new material synthesized by bacteria. Until now, *Lactobacillus*-derived CLNA has not been comprehensively studied, and there are no functional studies on any *Lactobacillus*-derived single isomers of CLNA.

We succeeded in separating the CLNA mixture and demonstrated that CLNA1 (c9, t11, c15-CLNA) and CLNA2 (c9, t11, t15-CLNA) inhibit the growth of three types of colon cancer cells in a time- and concentration-dependent manner. Our results and the lethal concentrations are similar to previous reports, which demonstrated that plant-derived CLNAs exerted potent anti-proliferation effects on a wide range of human and murine cancer cell lines in vitro [4,22,25]. In addition, the IC_50_ of ALA on Caco-2 cells is about two times that of CLNA1 and CLNA2, which suggests that CLNA isomers have a stronger inhibitory effect on Caco-2 cells, and that *Lactobacillus plantarum* ZS2058 was capable of producing a type of more toxic fatty acid.

The inhibitory mechanism of Caco-2 cells by *Lactobacillus*-derived CLNA was also unknown, so we investigated whether CLNAs could inhibit Caco-2 cells through triggering cell cycle arrest. Our results showed that CLNAs did not trigger cell cycle arrest. This is different from jacaric acid (c8, t10, c12-CLNA), which can trigger cell cycle arrest at the G0/G1 phase [22,25]. With Annexin-V/PI staining, we found that most Caco-2 cells were in the late stage of apoptosis or necrosis, which can be considered a characteristic of pyroptosis according to the Feng Shao’s study [26].

Pyroptosis can be induced by the canonical caspase-1 inflammasomes, or by the activation of caspase-4, -5 and -11 by cytosolic lipopolysaccharide [27]. The activated caspases cleave gasdermin D (GSDMD) at its middle linker domain to release autoinhibition on its gasdermin-N domain, which executes pyroptosis via its pore-forming activity [28,29,30]. Interestingly, CLNA1 and CLNA2 result in Caco-2 cell death by different pathways, although their structures are almost identical. It has been found that VX-765, a targeted inhibitor of caspase-1, can suppress the Caco-2 cell death induced by CLNA1 and CLNA2, and the broad-spectrum caspase inhibitor AC is also effective. Further Western blotting experiments showed that caspase-1 and GSDMD in Caco-2 cells stimulated by CLNA1 showed a cleavage band, but caspase-4 and caspase-5 did not show a cleavage band. However, Caco-2 cells treated with CLNA2 displayed caspase-4, caspase-5, and GSDMD cleavage bands, while caspase-1 did not show cleavage bands. Therefore, these results demonstrated that CLNA1 caused Caco-2 cell pyroptosis may through the caspase-1 mediated classical pathway, and that CLNA2 caused Caco-2 cell pyroptosis may through caspase-4 and caspase-5 mediated non-canonical pathways. These findings are in line with recent studies on the mechanism of DHA on cancer cells. DHA and CLNA both belong to omega-3 polyunsaturated fatty acids. Several studies have found that the DHA can trigger caspase-1 and GSDMD activation to cause pyroptosis in breast cancer cells, and enhance the secretion of IL-1β [31]. BV-2 microglial cells also undergo a proinflammatory cell death program like pyroptosis when treated with DHA [32].

We also found that lipid peroxidation may serve as an accelerator of pyroptosis in Caco-2 cells during treatment with CLNA1 and CLNA2. MDA can be regarded as an index of lipid peroxidation [33], and the lipid peroxidation in Caco-2 cells was enhanced during the treatment of CLNAs. This is consistent with previous reports of bifidobacterially produced CLNA and most plant-derived CLNAs on cancer cells [14,17,25]. In addition, the growth-inhibitory effects of *Lactobacillus*-derived CLNAs on Caco-2 cells were reduced upon the addition of the antioxidant VE. The antioxidant defense enzyme glutathione peroxidase 4 (GPX4) is a negative regulator of the pyroptotic cell death pathway, and plays an important role in inhibiting lethal inflammation associated with sepsis [34,35], which suggests that lipid peroxidation can drive GSDMD-mediated pyroptosis.

Numerous studies have confirmed that plant-derived CLNAs can cause apoptosis [25,36]. Jacaric acid induced cell death of LNCaP cells through activation of intrinsic and extrinsic apoptotic pathways, resulting in cleavage of PARP-1 and increased cleavage of caspase-3, -8, and -9 [5]. Punicic acid was also found to result in intrinsic apoptosis via a caspase-dependent pathway [36]. However, we found that the pan-caspase inhibitor Z-VAD had no effect on CLNA-induced Caco-2 cell death, and that apoptosis-related proteins caspase-3/7/9 and PARP-1 also did not show any cleavage products, so the inhibition of Caco-2 cells by *Lactobacillus*-derived CLNAs is not via apoptosis. These results confirm that there is a difference in inhibitory mechanisms between *Lactobacillus*-derived CLNAs and plant-derived CLNAs, which may be due to their structural differences.

In addition, we found that the necrotic apoptosis inhibitor NSA can significantly inhibit the toxic effects of CLNAs on Caco-2 cells. NSA inhibitors can bind to the GSDMD protein, making it impossible to expose the N-terminus and forming holes on the cell membrane leading to pyroptosis [37]. This can explain the result that NSA can inhibit the death of Caco-2 cells from CLNA. Therefore, there may be some interaction between necroptosis and pyroptosis induced by CLNA that must be further investigated. Unlike CLNA1, the RIPK1 inhibitor Nec-1s was significantly effective in inhibiting CLNA2-induced Caco-2 cell death, which indicates that the pyroptosis induced by CLNA2 was related to necroptosis.

In summary, the in vitro antiproliferative effects of two conjugated linolenic acid isomers derived from lactic acid bacteria were studied, and pyroptosis was confirmed as the inhibitory mechanism of Caco-2 cancer cells when treated with these microbially produced conjugated linolenic acid isomers. However, there were some differences in the inhibitory mechanism between CLNA1 isomer and CLNA2 isomer, which emphasized the importance of chemical structure in CLNAs.

## 4. Material and Methods

### 4.1. Microorganism Cultivation and Preparation of CLNA Mixture

*L. plantarum* ZS2058 were sub-cultured twice in de Man, Rogosa and Sharpe (mMRS) medium at 37 °C for 24 h, then inoculated into 2 L of mMRS with α-linolenic acid (Nu-chek Prep, Shanghai, China) at a final concentration of 30 mg/mL, and cultured for 48 h. After culturing, the fermentation broth was placed in a pear-shaped separatory funnel with the ratio of fermentation broth: isopropanol: *n*-hexane = 3:2:3 (*v*/*v*/*v*), shaken thoroughly, and allowed to stand for 15 min. The solution was layered, and the upper transparent *n*-hexane layer was collected. The *n*-hexane layer was then evaporated on a rotary evaporator and re-dissolved in methanol (chromatography grade) to the concentration of 40 mg/mL.

### 4.2. Separation of CLNA and Purity Detection

The crude extracts were separated by RP-HPLC on an Ultimate^®^ 5 μm C30 Semi-preparative column (10 × 250 mm) (Yuexu Technology, Shanghai, China) by a Waters 2545 RP-HPLC. The CLNA isomers were separated in methanol/water/formic acid (80:20:0.01, *v*/*v*) at a flow rate of 5 mL/min^−1^ with an injection volume of 700 μL. The CLNA fragments were detected by a UV detector at an absorbance of 205 nm and 233 nm, and fractions containing the single isomer were collected automatically when both absorbances were detected. The methanol and water from the pooled fractions were removed by rotary evaporation (37 °C), then the single CLNA isomer was re-extracted with methanol for storage. The purity of the extracted CLNA1 (cis9,trans11,cis15-CLNA) and CLNA2 (trans9,trans11,cis15-CLNA) was determined by GC-MS as previously described [38].

### 4.3. Cell Culture

Caco-2 colon cancer cells (Shanghai Institute of Cellular Sciences, Chinese Academy of Sciences, China) were cultured in DMEM high-glucose complete medium supplemented with 5% fetal calf serum (Gibco company, New York, NY, USA), 1% sodium pyruvate, 1% non-essential amino acids, and 1% penicillin. SW480 and HT-29 colon cancer cells (Shanghai Institute of Cellular Sciences, Chinese Academy of Sciences, Shanghai, China) were cultured in RPMI 1640 complete medium supplemented with 5% fetal calf serum, and 1% sodium pyruvate. Caco-2 cells were seeded onto a 96-well plate (Fisher, Ottawa, ON, Canada) at a concentration of 8000 cells/well in 100 μL complete culture medium. Similarly, SW480 and HT-29 cells were seeded at a concentration of 5000 cells/well or 3000 cells/well, respectively. The concentration of Caco-2 cells was adjusted to 4 × 10^5^ cells/well when the cells were grown in 6-well plates with 2 mL complete medium. All of the media and supplements for tissue culture were purchased commercially (Gibco, New York, NY, USA). All cells were maintained in a 37°C humidified incubator with 5% CO_2_.

### 4.4. MTT Cell Viability Assay

The 1% FBS medium with CLNA concentrations of 0 μM, 10 μM, 20 μM, 30 μM, 40 μM, and 50 μM were added to a 96-well plate and cultured for 48 h. At the end of treatment, culture media were removed and 3-(4,5-dimethylthiazol-2-yl)-2,5-diphenyltetrazolium bromide (MTT) (Sigma-Aldrich, Saint Louis, USA) was added to the cells at a concentration of 0.5 mg/mL. After 4 h incubation at 37°C, the MTT reagent was removed and DMSO was used to solubilize the formazan dye formed by viable cells.

### 4.5. Cell Growth Inhibition Assay

Caco-2 cells seeded on a 96-well plate were stimulated by inhibitors added as follows. After treatment Monmouthfor 48 h, the MTT experiment was carried out to determine the cell viability. The working concentrations of Vitamin E, Z-VAD-FMK, 3-Methyladenine, VX-765, GSK-872 (MedChemExpress, Monmouth, NJ, USA), Necrosulfonamide and Nec-1s (Selleck, Houston Texas USA) were 30, 10, 10, 5, 10, 0.5, and 10 μM, respectively. The inhibitors were dissolved in DMEM (1% BSA + 0.1% DMSO) and combined with the CLNAs at their IC_50_.

### 4.6. PI Staining Assay

Cells were exposed for 48 h in a 6-well plate to solvent (negative) control medium containing 1% FBS, 0.1% DMSO, and CLNA isomers at half the IC_50_ or the IC_50_ concentrations. Cells were collected for fixation and stored at 4  °C prior to cell cycle analysis. The cells were then harvested with 70% ethanol, centrifuged, washed twice with PBS and then stained with a solution containing 100 μg/mL RNase A, 0.2% Triton X-100 and 50 μg/mL propidium iodide (Sigma-Aldrich, St. Louis, MO, USA) for 1 h at 4 °C in darkness before the cell cycle was measured with a flow cytometer within 1 h [39].

### 4.7. Annexin-V/PI Staining Assay

Cells were exposed for 48 h in a 6-well plate to medium containing 1% FBS, 0.1% DMSO, and CLNA isomers at either half the IC_50_, the IC_50_, and two times the IC_50_ concentration, respectively. The Annexin V-FITC/propidium iodide (PI) apoptosis detection kit was purchased from BD Pharmingen (San Diego, CA, USA), and cell aliquots containing 1 × 10^5^ cells in 100 μL buffer were stained with 5 μL PI solution and 5 μL FITC-conjugated Annexin V for 15 min at 37  °C. After staining, 400  μL Binding buffer was added to the cells, and samples were stored on ice until data acquisition [39]. All analyses were performed using the software FlowJo_V10.

### 4.8. Western Blotting Analysis

Proteins were extracted using Lysis Buffer (Biyuntian, Shanghai, China) and Protease and Phosphatase Inhibitor (Biyuntian, Shanghai, China). The protein concentrations were determined by the bicinchoninic acid (BCA) method (Biyuntian, Shanghai, China). Then, 12% polyacrylamide gels were transferred to PVDF membranes using a semi-dry system. The membrane was blocked for 1 h and incubated overnight at 4 °C with a primary antibody (Anti-caspase-1/3/4/5/7/9, anti-cleaved N-terminal GSDMD, Abcam, Cambrige, UK). The membrane was then incubated for 2 h with a secondary antibody (Jackson ImmunoResearch). The anti-β-actin antibody (ZSGB-BIO, Beijing, China) was used as a loading control. The bands were analyzed using Alpha View software (Version 3.4.0) for densitometry analysis.

### 4.9. RNA Extraction and RT-qPCR

Cells were collected in the logarithmic growth phase. After washing twice with PBS buffer, 500 μL of prechilled TRIZOL was added to each well. Cells were incubated on ice for 20 min and scraped into an enzyme-free tube using a cell scraper. RNA extraction and reverse transcription were carried out as described previously [40]. GAPDH was used as the reference gene for real-time fluorescence quantitative PCR analysis, and the primer sequences are shown in Table 2.

### 4.10. Statistical Analysis

All data are presented as mean ± SEM, and experiments were performed at least in triplicate before analysis with GraphPad Prism 6 statistical software (GraphPad Software, Inc, San Diego, CA, USA). An independent t-test was used to determine significant differences using SPSS 19.0 software (SPSS Inc, Chicago, IL, USA) or GraphPad Prism 6 (GraphPad Software, Inc, San Diego, CA, USA). A *p* value <0.05 was considered statistically significant.

## Figures and Tables

**Figure 1 molecules-25-01225-f001:**
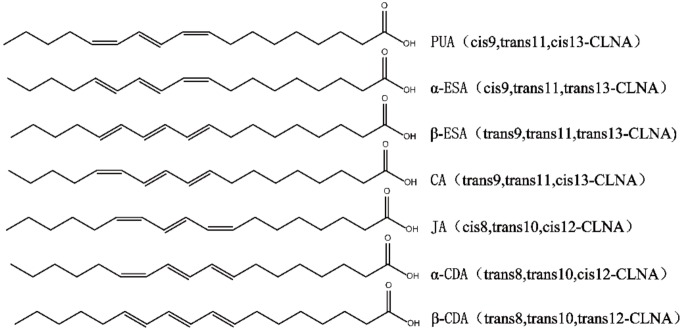
Chemical structure of plant-derived CLNAs.

**Figure 2 molecules-25-01225-f002:**
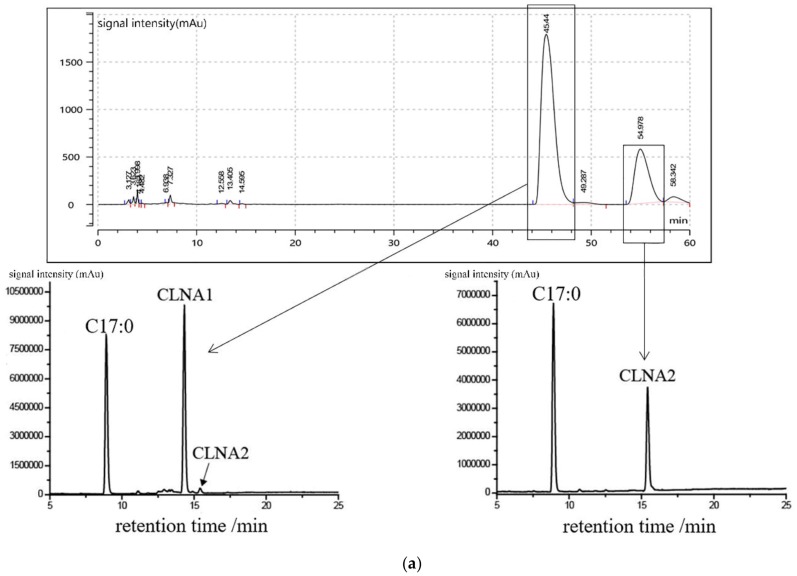
Preparation and purity detection of CLNA1 and CLNA2. (**a**) Separation of CLNA1 and CLNA2 by liquid chromatograph. The purity of CLNA1 and CLNA2 detected by GC-MS (C17:0 was added as a standard). (**b**) Mass spectrum and chemical structures of conjugated linolenic acid (CLNA) isomers used in this study (CLNA1: c9, t11, c15-CLNA, CLNA2: t9, t11, c15-CLNA).

**Figure 3 molecules-25-01225-f003:**
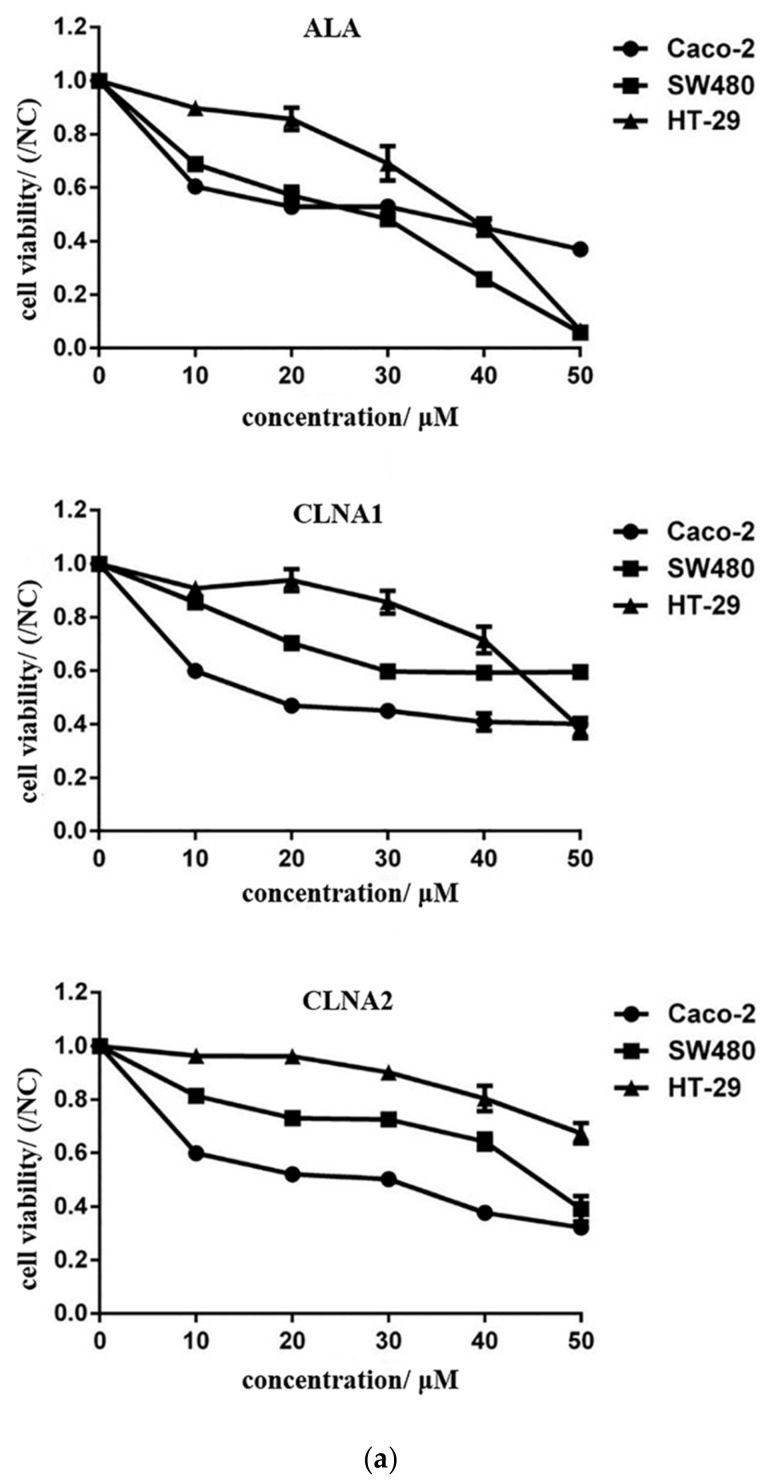
Anti-proliferative activity of CLNA1, CLNA2 against colon cancer cells. (**a**) Anti-proliferative effects of ALA, CLNA1, CLNA2 on three kinds of colon cancer cell lines and the corresponding IC50. (**b**) Morphology and quantity of Caco-2 cells treated by CLNA1 and CLNA2 at different time points. And the scale bar of every picture is 50μm. (**c**) Flow cytometry of propidium iodide and annexin V-fluorescein isothiocyanate (FITC)-stained Caco-2 cells stimulated by different concentrations of CLNA1 or CLNA2 for 48 h. Data are presented as mean ± SEM and at least three separate experiments were performed in all studies. *, *p* < 0.05, **, *p* < 0.01 and ***, *p* < 0.001 compared with control.

**Figure 4 molecules-25-01225-f004:**
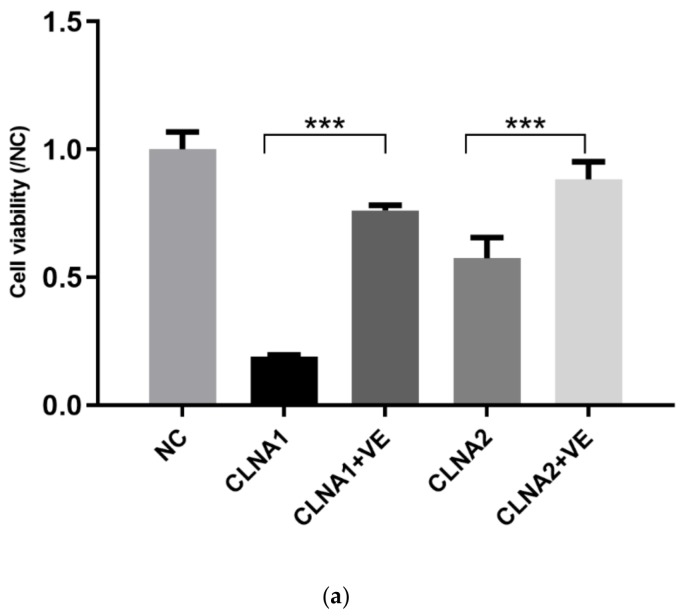
Induction of lipid peroxidation by CLNA1 and CLNA2 in Caco-2 cells. (**a**) The effects of inhibitor VE on the Caco-2 cells treated with CLNA1 and CLNA2, respectively (*n* = 6). (**b**) The changes of MDA protein expression in Caco-2 cells during the treatment with CLNA isomers (*n* = 3). NC indicated control group without CLNA treatment. Data are presented as mean ± SEM and at least three separate experiments in all studies. * *p* < 0.05, ** *p* < 0.01 and *** *p* < 0.001 compared with control.

**Figure 5 molecules-25-01225-f005:**
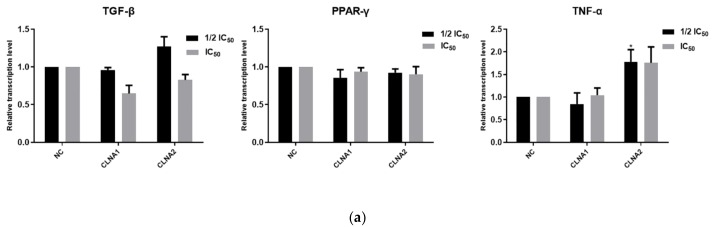
CLNA1 and CLNA2 induce Caco-2 cell death independent of apoptosis and autophagy. (**a**) The effects of CLNA1 and CLNA2 on relative transcription levels of apoptosis-related factors PPARγ, TNF-α and TGF-β in Caco-2 cells. (**b**) The effects of CLNA1 and CLNA2 on expression of apoptosis proteins PARP-1, caspase-3, caspase-7 and caspase-9 in Caco-2 Cells. (**c**) Ratio of Caco-2 cells at each stage of different cell cycles after treatment with CLNA1 and CLNA2 under different concentrations for 48 h. (**d**) The effects of different inhibitors on Caco-2 cells treated by CLNA1 and CLNA2 (*n* = 6). NC indicated control group without CLNA treatment. Data are presented as mean ± SEM and at least three separate experiments were performed in all studies. * *p* < 0.05, ** *p* < 0.01 and *** *p* < 0.001 compared with control.

**Figure 6 molecules-25-01225-f006:**
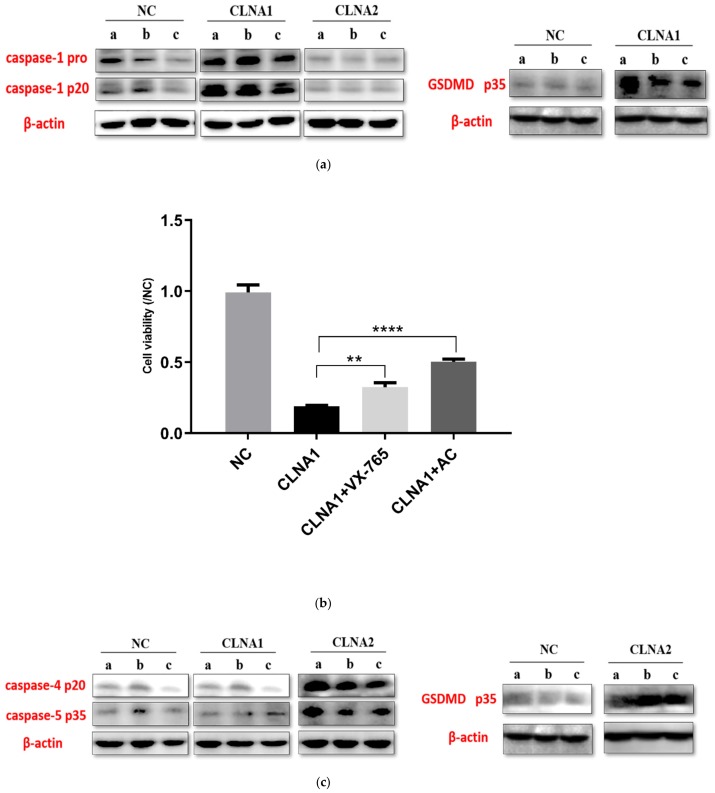
CLNA1 and CLNA2 induce Caco-2 pyroptosis by different pathways. (**a**, **c**) The expression of proteins associated with pyroptosis in colon cancer cells treated by CLNA1 or CLNA2. (**b**) The effects of caspase-1 inhibitor VX-765 and lipid peroxdation inhibitor VE on Caco-2 cells treated by CLNA1. (**d**) The effects of caspase-4/5 inhibitor AC and lipid peroxdation inhibitor VE on Caco-2 cells treated by CLNA2. a, b, and c indicate parallel tests. NC indicated control group without CLNA treatment. Data are presented as mean ± SEM and at least three separate experiments were performed in all studies. * *p* < 0.05, ** *p* < 0.01, and *** *p* < 0.001 compared with control.

**Table 1 molecules-25-01225-t001:** The IC_50_ of ALA, CLNA1, and CLNA2 on three colon cancer cells.

Cell	ALA	CLNA1	CLNA2
Caco-2	38.93 ± 2.18	18.26 ± 2.06	19.75 ± 1.83
SW480	21.08 ± 0.67	59.46 ± 4.67	43.81 ± 11.81
HT-29	34.46 ± 1.31	47.07 ± 8.22	67.52 ± 2.87

**Table 2 molecules-25-01225-t002:** Primers for RT-qPCR.

Primers	Sequence (5′–3′)
GAPDH-F	CCTGGCCAAGGTCATCCATG
GAPDH-R	GGAAGGCCATGCCATGGAGC
PPARγ-F	ATGGAGCCCAAGTTTGAGTTT
PPARγ-R	TGTCTGAGGTCCGTCATTTTC
TNF-α-F	ATGAGCACAGAAAGCATGATC
TNF-α-R	TACAGGCTTGTCACTCGAATT
TGF-β-F	GGCCAGATCCTGTCCAAGC
TGF-β-R	GTGGGTTTCCACCATTAGCAC

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
