# Peer review of "Antiproliferation Activity and Mechanism of c9, t11, c15-CLNA and t9, t11, c15-CLNA from Lactobacillus plantarum ZS2058 on Colon Cancer Cells"

_molecules, 2020, doi:10.3390/molecules25051225_

Round 1

Reviewer 1 Report

This manuscript describes the antiproliferation activity of conjugated linolenic acid on colon cancer cells and its mechanism. The authors performed a plenty of experiments. The experimental data are promising. Thus, the reviewer recommend this manuscript for publication in Molecules.

The reviewer found some mistakes in the manuscript. Please correct them.

  1. In Figure 5 (a): Please show the asterisks which show significant difference.
  2. References: Please follow the author guidelines how to write references. The reviewer found that seome references are written in abbreviated style and other ones are written full titile of journal's name.

Author Response

Reviewer 1:

This manuscript describes the antiproliferation activity of conjugated linolenic acid on colon cancer cells and its mechanism. The authors performed a plenty of experiments. The experimental data are promising. Thus, the reviewer recommend this manuscript for publication in Molecules.

The reviewer found some mistakes in the manuscript. Please correct them.

Comment1:In Figure 5 (a): Please show the asterisks which show significant difference.

Response1: We are sorry for the mistake, and the significance has been analyzed and added in the new Figure.5a.

Comment2:References: Please follow the author guidelines how to write references. The reviewer found that seome references are written in abbreviated style and other ones are written full titile of journal's name.

Response2: We are sorry for the mistakes, the format of references has been revised seriously.

Reviewer 2 Report

Comments to the author:

The authors described the isolation and biological characterization of Lactobacillus-derived CLNA. And the possible pathway of Lactobacillus-derived CLNA isomer on colon cancer cells were proved. Overall, the study is well thought out. Some comments below.

  1. Conjugated linolenic acid (CLNA) is a type of ω-3 fatty acid that shows pleiotropic mechanism of action in cells. The authors showed antiproliferation data on colon cancer cells. Would like to see the toxicity on normal non-cancerous cells.

  1. Why the authors chose colon cancer cell instead of other cancer cell lines? Interested to see any rationale behind this?Additionally, it would be better to show Sigmoid curve of cmpds effect on cell proliferation with IC50 values shown in the Figure for clarity.

  1. The writing is sloppy and need to be improved before acceptance for publication.

i.e. Page 2 Line 54 Figure.1. should be Figure 1.

Page 14 Line 372 Table legend should be uniformed.

Figure 5 IC50 should be IC50.

Figure 5d is missing.

Author Response

Reviewer 2:

The authors described the isolation and biological characterization of Lactobacillus-derived CLNA. And the possible pathway of Lactobacillus-derived CLNA isomer on colon cancer cells were proved. Overall, the study is well thought out. Some comments below.

Comment1:Conjugated linolenic acid (CLNA) is a type of ω-3 fatty acid that shows pleiotropic mechanism of action in cells. The authors showed antiproliferation data on colon cancer cells. Would like to see the toxicity on normal non-cancerous cells.

Response1: Thanks for the question. In previous in vitro experiment, jacaric acid (c8, t10, c12-CLNA,a typical plant-derived CLNA) on the viability of normal murine cells have been reported. Jacaric acid exhibited little direct cytotoxicity on normal primary myeloid cells such as murine bone marrow cells, peritoneal macrophages, splenocytes and thymocytes, as the percentage of cell viability of the cells remained >80 % even when the cells were incubated with 100 µM jacaric acid for 48 h. (Jacaric acid inhibits the growth of murine macrophage-like leukemia PU5-1.8 cells by inducing cell cycle arrest and apoptosis. Cancer Cell Int 2015, 15, 90).

Although there were few known about bacterial CLNA, based on the previous results of different CLNAs, we presume the toxicity of bacterial CLNA on normal non-cancerous cells are limited.

Comment2:Why the authors chose colon cancer cell instead of other cancer cell lines? Interested to see any rationale behind this? Additionally, it would be better to show Sigmoid curve of cmpds effect on cell proliferation with IC50 values shown in the Figure for clarity.

Response2: There are many researches on different plant-derived CLNAs, which were carried out on colon cancer in vivo and in vitro. As the bacterial CLNA had few researches, we purified each isomer and carried out the current work to explore the effect of bacterial CLNA on colon cancer.

(Here a image can't show, please check it the attachment. )

We are sorry for the incomplete information about IC50, and the accurate IC50 was provided in the listed table. It might be difficult to mark the accurate IC50 exactly on the Sigmoid curve  and a table may be better to show the IC50.

The IC50 values of ALA, CLNA1, and CLNA2 on three colon cancer cells

Cell

ALA

CLNA1

CLNA2

Caco-2

38.93±2.18

18.26±2.06

19.75±1.83

SW480

21.08±0.67

59.46±4.67

43.81±11.81

HT-29

34.46±1.31

47.07±8.22

67.52±2.87

Comment3:The writing is sloppy and need to be improved before acceptance for publication.

i.e. Page 2 Line 54 Figure.1. should be Figure 1.

Page 14 Line 372 Table legend should be uniformed.

Figure 5 IC50 should be IC50.

Figure 5d is missing.

Response3: We were really sorry for the careless mistakes. Thank you for your reminding. We have revised the manuscript and some mistakes marked yellow were also corrected.
